# The antidiabetic drug metformin aids bacteria in hijacking vitamin B12 from the environment through RcdA

Luxia Yao[1,2,3,4,5], Yihan Wang[2,3,4,5], Shenlu Qin[2,3,4], Shihao Zhu[2,3,4] & Lianfeng Wu [2,3,4 ✉]

Years of use of the antidiabetic drug metformin has long been associated with the risk of vitamin B12 (B12) deficiency in type 2 diabetes (T2D) patients, although the underlying mechanisms are unclear. Accumulating evidence has shown that metformin may exert beneficial effects by altering the metabolism of the gut microbiota, but whether it induces human B12 deficiency via modulation of bacterial activity remains poorly understood. Here, we show that both metformin and the other biguanide drug phenformin markedly elevate the accumulation of B12 in *E. coli*. By functional and genomic analysis, we demonstrate that both biguanides can significantly increase the expression of B12 transporter genes, and depletions of vital ones, such as *tonB*, nearly completely abolish the drugs' effect on bacterial B12 accumulation. Via high-throughput screens in *E. coli* and *C. elegans*, we reveal that the TetR-type transcription factor RcdA is required for biguanide-mediated promotion of B12 accumulation and the expressions of B12 transporter genes in bacteria. Together, our study unveils that the antidiabetic drug metformin helps bacteria gather B12 from the environment by increasing the expressions of B12 transporter genes in an RcdA-dependent manner, which may theoretically reduce the B12 supply to T2D patients taking the drug over time.

[1] Fudan University, Shanghai, China. [2] Key Laboratory of Growth Regulation and Translational Research of Zhejiang Province, School of Life Sciences, Westlake University, Hangzhou, Zhejiang, China. [3] Westlake Laboratory of Life Sciences and Biomedicine, Hangzhou, Zhejiang, China. [4] Institute of Basic Medical Sciences, Westlake Institute for Advanced Study, Hangzhou, Zhejiang, China. [5]These authors contributed equally: Luxia Yao, Yihan Wang. ✉email: wulianfeng@westlake.edu.cn

Currently, metformin is still the first-line pharmaceutical option for treating type 2 diabetes mellitus (T2D) worldwide[1,2]. However, despite many beneficial effects proposed for the use of metformin, it is found that long-term use of the drug commonly results in B12 deficiency in T2D patients, with a prevalence up to 41%[3–5]. Individuals with B12 deficiency usually exhibit neuropathy with symptoms, such as ataxia, diminished cognition, hematologic abnormalities of microcytosis and megaloblastic anemia[6]. Specifically, metformin-induced low B12 status has been linked with a reduction in cognitive function and an increase in the risk of depression, which are generally found to be irreversible upon the occurrence of demyelination and nerve degeneration[7–9].

B12 is indispensable for cellular growth and homeostasis due to its essential roles in DNA synthesis and metabolism of amino acids as well as fatty acids[10]. Humans and the majority of small-intestinal microbes are completely dependent on a dietary source of B12, as they lack B12 biosynthesis genes[11,12]. Dietary B12 is initially released in the stomach, where it binds to intrinsic factor to form the complex followed by binding with the cubilin receptor on the distal ileum, eventually being sent to the liver for storage[13]. The amount of B12 stored in the liver is usually sufficient to meet physiological needs, which has made difficult to experimentally induce B12-deficient models for mechanistic studies[14,15]. Likely, at least in part, due to the lack of sound B12-deficient models, the mechanisms underlying metformin-induced B12 deficiency remain largely unknown.

Numerous lines of evidence suggest that metformin exerts its beneficial effects, including antihyperglycemic and antiaging effects, by altering the gut microbiome[16–20]. Intriguingly, it has been reported that biguanide-associated B12 deficiency can be reversed by the administration of antibiotics[21], indicating a potential role of bacteria in biguanide-induced B12 malabsorption in humans. It is noted that the uptake of metformin and B12 in the host occurs through the distal ileum, where a considerable amount of B12 auxotroph gut microbes exist and provides a platform for potential interactions among the drug, B12 and microbes. A recent study also indicated that bacterial B12 transporters may help the microbiota compete with the host for B12 by dissociating the host intrinsic factor-B12 complex[22]. Moreover, the depletion of B12 by the gut microbiota has been proposed as a primary cause of B12 malabsorption in certain human disorders, such as blind loop syndrome[23,24]. However, whether metformin influences the host B12 level through gut microbes remains unknown.

Unlike the symbiotic relationship between humans and gut microbes, *C. elegans* directly eats bacteria as a source of food, making it a suitable tool to detect B12 levels in bacteria[25,26]. In the present study, through functional analysis of B12 activity in worms and direct measurement of bacterial B12 levels, we demonstrated that biguanides significantly increased the amount of B12 in multiple *E. coli* strains. Interestingly, such B12 accumulation induced by biguanides was found to be fully dependent on the activity of functional B12 transporters. Mechanistically, we found that both biguanides could increase the expressions of certain vital B12 transporter genes in *E. coli*, especially *tonB*, suggesting a possible mechanism by which biguanides promote B12 accumulation in bacteria. Through a genome-wide screen with the *E. coli* Keio collection library for genes responsible for biguanide action using B12-sensing worm models[25–27], among which the *tonB* mutant bacterial clone was included as a positive control, we revealed that the biofilm formation process and transmembrane transport were the top enriched pathways. Knockout of genes from those enriched pathways significantly impaired the effect of phenformin in elevating bacterial B12 levels. By conducting another round of narrow-down filtering, we further identified the transcription factor RcdA, a TetR-type transcription factor[28,29], as the element responsible for biguanide-increased B12 accumulation in *E. coli*. Moreover, we demonstrated that RcdA was required for the biguanide-mediated increase in the expressions of corresponding B12 transporter genes, indicating that RcdA acts upstream of B12 transporters in response to metformin exposure. Altogether, our study not only provides insights into the mechanisms underlying the effect of biguanide drugs in helping bacteria capture B12 from the environment but also lays a foundation for understanding the role of the gut microbiota in B12 deficiency induced by long-term use of metformin in T2D patients.

## Results

**Metformin and phenformin promote bacterial B12 accumulation.** To investigate whether biguanides can change the level of B12 in bacteria, we employed the *nhr-114* loss-of-function (hereafter abbreviated as *Δnhr-114*) worm model, which is a newly proposed B12-deficient model developed by us and others[25,27]. Phenotypically, these animals exhibit fertility in a B12 dose-dependent manner. Interestingly, we found that the infertility-rescuing effect of biguanides was completely dependent on the presence of B12, which disappeared when drugs were supplied under B12-deficient conditions (Fig. 1a). To determine whether biguanides elevate the levels of B12 in worms through bacteria, we used the well-established B12 sensor *Pacdh-1::GFP* worms, whose GFP intensity can indicate the level of B12 present in the diet[26,30,31]. We administered phenformin to *Pacdh-1::GFP* worms in an axenic culture system (Fig. 1b–d) or fed the sensor worms with drug-pretreated *E. coli* BW25113 (Fig. 1e, f), another widely-used *E. coli* strain for mechanistic studies[32–34]. Surprisingly, we found that biguanides increased the levels of B12 in the B12 sensor worms only when the drugs were administered in the presence of bacteria, suggesting that biguanides promoted B12 accumulation in bacteria rather than worms directly. To possibly mimic the physiological condition in patients where bacteria are exposed to a long-term treatment with low-dose biguanides, we explored whether a lower dose of phenformin could induce bacterial B12 accumulation in a relatively longer treatment compared to the conditions mainly applied in our study. Indeed, we found that B12 levels in bacteria were increased by 2 mM phenformin with time (Supplementary Fig. 1), suggesting that such bacterial B12 absorption increase may occur in T2D patients during the long-term and low-dose therapy with metformin.

To examine the exact extent to which the B12 levels were changed by biguanides in bacteria, we employed liquid chromatography with tandem mass spectrometry (LC-MS/MS) to measure B12 levels in bacteria with or without drug treatment. The results showed that both drugs significantly increased the accumulation of B12 in the adenosylcobalamin (Ado-Cbl) and cyanocobalamin (CN-Cbl) forms in bacteria (Fig. 1g). Theoretically, the CN-Cbl form is the exogenously supplemented version of B12, while the Ado-Cbl form should be converted from CN-Cbl inside of live bacteria[35,36]. Interestingly, we found that phenformin at 4 mM induced a much higher accumulation of CN-Cbl but a lower level of Ado-Cbl compared to those outcomes induced by metformin at 200 mM (Fig. 1g), which may indicate a more potent activity of phenformin in blocking the bacterial conversion of Ado-Cbl from CN-Cbl than metformin as reported in other applications[37,38]. Together, these results demonstrated that biguanides promote bacterial B12 accumulation from the environment.

**The B12 transport system is required for phenformin-induced bacterial B12 accumulation.** We administered phenformin with

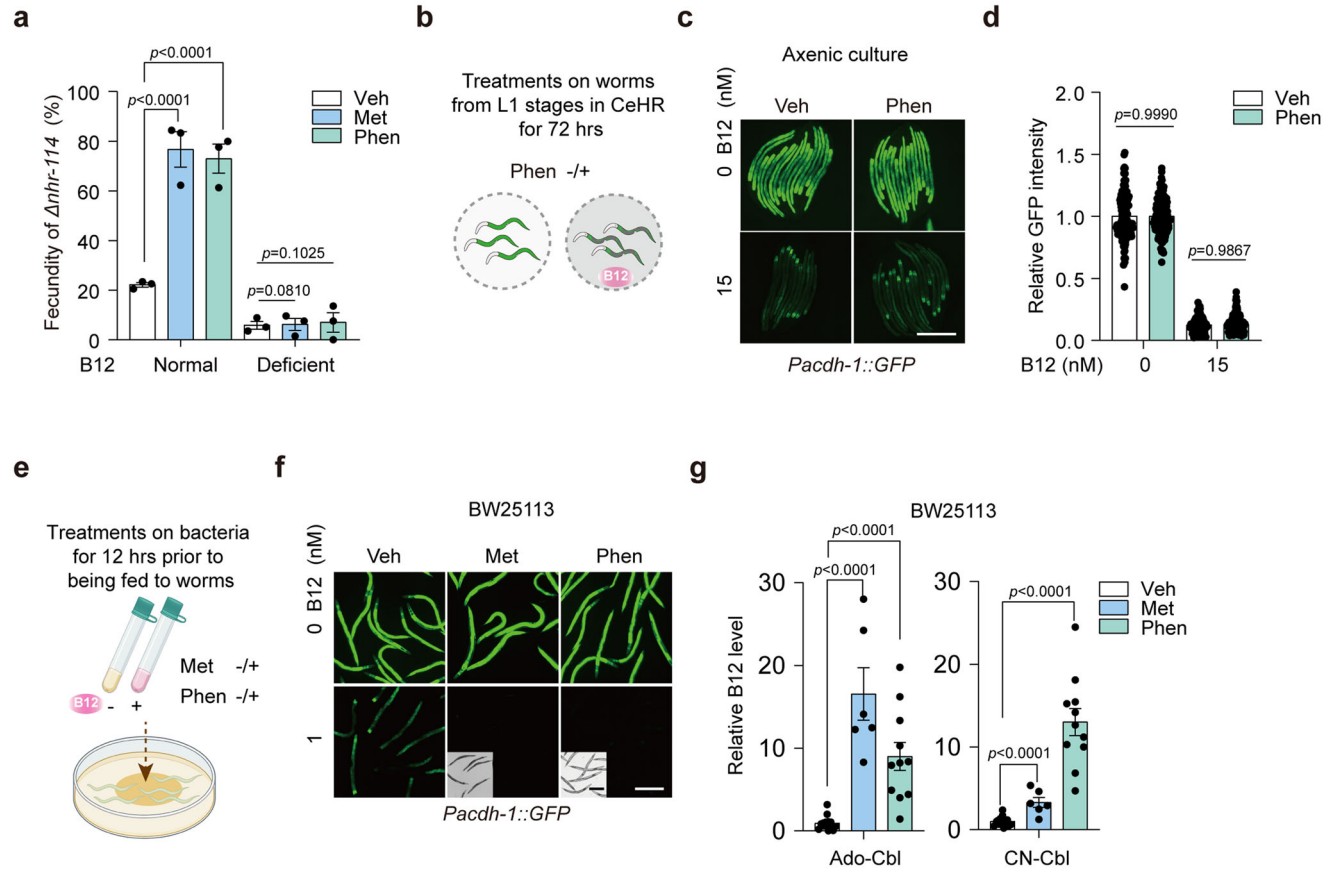

**Fig. 1 The antidiabetic biguanides promote bacterial B12 accumulation. a** Fecundity of Δ*nhr-114* animals on normal NGM plates or B12-deficient NGM plates treated with 50 mM metformin/5 mM phenformin. Veh, vehicle. Met, metformin. Phen, phenformin. *N* = 3 independent experiments containing at least 30 worms per group. **b** Scheme of treatments on worms in axenic culture treated with B12 and/or phenformin. **c** Representative images of *Pacdh-1::GFP* worms supplemented with B12 and/or 4 mM phenformin in axenic culture. **d** Quantification of relative GFP intensity of *Pacdh-1::GFP* worms in (**c**). *N* = 3 independent experiments containing at least 30 worms per condition. **e** Scheme of treatments on BW25113 to feed *Pacdh-1::GFP* worms. **f** Representative images of *Pacdh-1::GFP* worms fed with BW25113 pretreated with B12 and/or 200 mM metformin/4 mM phenformin. *N* = 3 independent experiments containing at least 30 worms per condition. Scale bar: 250 μm (fluorescence images) and 500 μm (bright field images) for (**c**) and (**f**). **g** LC-MS/MS measurement of B12 levels in BW25113 treated with 200 mM metformin/4 mM phenformin. Ado-Cbl, adenosylcobalamin; CN-Cbl, cyanocobalamin. *N* = 2 independent experiments for the metformin groups and *N* = 3 independent experiments with at least 3 single colonies per test. The statistical significance values were determined by ordinary one-way ANOVA for (**a**) and (**d**) and unpaired *t* test for (**g**). Error bars denoted the S.E.M.

concomitant B12 or not to the *E. coli* strains that are typically used in *C. elegans* studies, including OP50, HT115, and HB101, to explore whether the drug could elevate the accumulation of B12 in different bacterial types. The treated bacteria were then examined by feeding to the B12 sensor worms and using LC-MS/MS measurement. Indeed, we found that phenformin strongly induced B12 accumulation in the K12 x B hybrid strain HB101 and the K12 strain HT115 (Fig. 2a and Supplementary Fig. 2a). However, it appeared that the B strain OP50 showed a weaker capacity to accumulate B12 from the environment than other strains either treated by phenformin or not, consistently indicated by the GFP intensities of B12 sensor animals and LC-MS/MS measurement on B12 levels (Fig. 2a and Supplementary Fig. 2a).

It has been documented that *E. coli* requires an active transport system to absorb B12 from the environment for growth and survival[39,40]. B12 is normally transported across the outer membrane by BtuB into the periplasm, which is dependent on the energy transduced from the TonB complex, composed of TonB, ExbB, and ExbD. In the periplasm, B12 is captured and delivered by BtuF to the ATP-binding-cassette transporter BtuCD and ultimately released into the cytoplasm (Fig. 2b). Thus, we speculated that the weak B12 accumulation capacity in the OP50 strain may be attributed to the low activity or expression of

genes in the control of B12 transport. Intriguingly, we found that the OP50 strain showed significantly lower expression of all the tested genes for B12 transport than BW25113 or HT115 (Fig. 2c), which explained its poor performance in absorbing B12 compared to that of other strains and the weakened effect of phenformin (Fig. 2a and Supplementary Fig. 2a). Considering the strong association between the activity of B12 transporters and the intrinsic capacity of bacterial B12 absorption, we wondered whether the B12 transport gene expression was altered by phenformin treatment. Indeed, we found that both metformin and phenformin induced the expressions of most of the B12 transporter genes, among which *tonB* was the most significant one (Fig. 2d). Furthermore, we confirmed that the deletion of *tonB*, *btuB* and *btuF* could markedly abolish the effect of phenformin on the induction of B12 accumulation in BW25113 (Fig. 2e), although not all the tested genes were from the B12 transport system (Supplementary Fig. 2b). Collectively, our results suggested that the B12 transport system, likely via its gatekeeping function, is essential for phenformin-induced B12 levels in *E. coli*.

**High-throughput screens for genes mediating the action of biguanides in bacteria.** To identify the genetic machinery

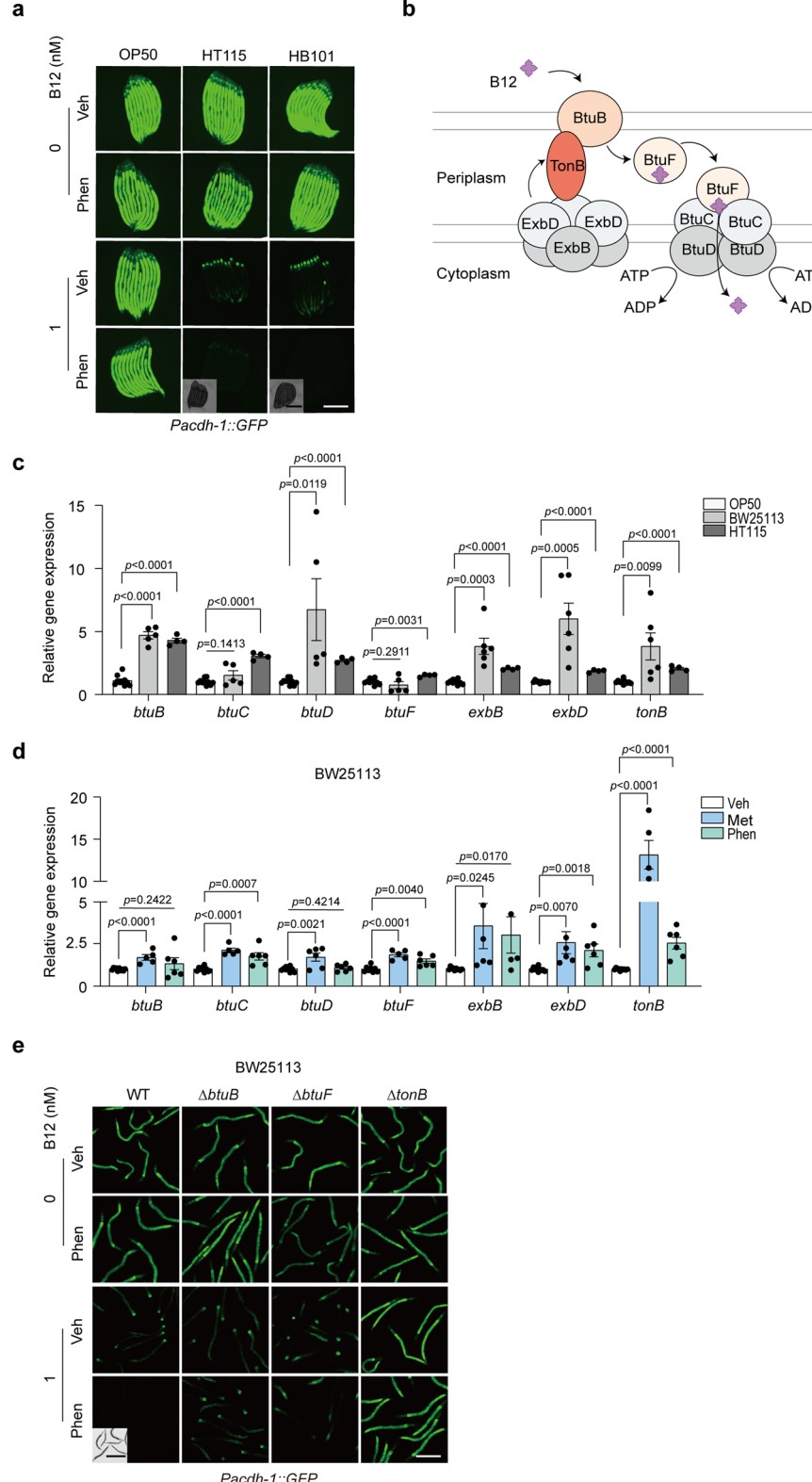

**Fig. 2 The B12 transport machinery plays essential roles in phenformin-mediated bacterial B12 accumulation. a** Representative images of *Pacdh-1::GFP* worms fed bacterial strains OP50, HT115, and HB101 pretreated with B12 and/or 4 mM phenformin. *N* = 3 independent experiments containing at least 30 worms per group. **b** Diagram of the B12 transport machinery in *E. coli*. **c**–**d** Relative mRNA levels of genes involved in B12 transport in OP50, BW25113, and HT115 bacteria (**c**) and BW25113 with 200 mM metformin/4 mM phenformin treatment or not (**d**). For (**c**, **d**), *N* = 2 independent experiments containing 4–6 replicates. The statistical significance values were determined by multiple *t* tests. Error bars denoted the S.E.M. **e** Representative images of *Pacdh-1::GFP* worms fed WT and B12 transporter mutant strains pretreated with B12 and/or 4 mM phenformin. WT, BW25113. *N* = 3 independent experiments containing at least 30 worms per group. Scale bar: 250 μm (fluorescence images) and 500 μm (bright field images) for (**a**) and (**e**).

responsible for bacterial B12 accumulation by biguanides, we set up genetic screens using Δnhr-114 animals whose reproduction could be fully restored by B12[25,27] and the B12 sensor worm strain Pacdh-1::GFP. To better design the screens, we compared the sensitivities of these two worm models upon supplementation with various doses of B12 and found that the Δnhr-114 readout showed higher sensitivity to the low dose of B12 than the GFP sensor strain, especially when 0.05 nM B12 was co-administered with phenformin to the bacteria (Fig. 3a, b). To evaluate the effect of bacterial mutants on worm reproduction, we included the tonB mutant strain as a positive control, in which basal B12 absorption and phenformin-mediated B12 induction were both found to be impaired (Fig. 2e). We confirmed that the effect of phenformin in inducing B12 accumulation in bacteria was significantly reduced in ΔtonB according to the low fecundity of Δnhr-114 (Fig. 3c and Supplementary Fig. 3a). Therefore, we carried out the primary unbiased screen with 3,985 bacterial mutants starting with Δnhr-114 fecundity readout (Fig. 3d and Supplementary Fig. 3b), followed by a secondary screen employing the B12 sensor animals Pacdh-1::GFP to narrow down the hits (Fig. 3e). From the primary screen, we identified 274 candidate genes, the knockout of which significantly reduced the effect of phenformin on bacterial B12 accumulation (Supplementary Data 1). Gene Ontology analysis revealed that biofilm formation and transmembrane transport were the most significantly enriched biological processes (Fig. 3f and Supplementary Data 1). From the secondary screen, 28 candidate genes were screened out, among which ΔtonB obtained the highest score (Fig. 3g and Supplementary Fig. 3c).

**Biguanides induce both bacterial growth inhibition and B12 accumulation by modulating RcdA.** To identify the target genes of biguanides that play the most consistent roles in the drug actions from the 28 hits, we included another trait of biguanides, bacterial growth inhibition (Fig. 4a), the phenotype of which has been considered to be closely associated with other effects of biguanides[41]. We first validated the growth inhibition effects of phenformin in four different E. coli strains (Supplementary Fig. 4), and then explored if such effects were bacteriostatic or bactericidal. The results showed that 5 mM phenformin treatment retarded the growth of E. coli, and slightly reduced the number of living cells, indicating that phenformin inhibited bacterial growth but did not kill them (Supplementary Fig. 5a). Moreover, 5 mM phenformin indeed induced dual effects on bacterial growth inhibition according to their cell density and integrity indicated by OD600 measurement and dsDNA binding dye SYBR Green I staining, respectively (Supplementary Fig. 5b, c). Collectively, we concluded that the growth inhibition effect of phenformin was mainly bacteriostatic.

We reasoned that deletion of the drug target genes would confer the mutant strains resistance to high concentrations of phenformin. Interestingly, only 3 out of the 28 candidates could grow up under phenformin treatment, among which ΔrcdA exhibited the most significant resistance to the drug (Fig. 4b). Moreover, RcdA is a transcription factor controlling the expression of many important genes involved in biofilm formation[28], which were candidates out of initial screens (Fig. 3f), suggesting that RcdA might be a key factor determining the effect of biguanides in inducing bacterial B12 accumulation. To test whether RcdA mediated the effects of phenformin by modulating drug accumulation, we measured the drug levels in ΔrcdA using LC-MS/MS. The results showed that the deletion of rcdA did not alter the amount of phenformin accumulated in bacteria compared to that in WT (Supplementary Fig. 6), indicating that RcdA mediated the activity of the drug instead of its level in bacteria.

To confirm the dual role of RcdA in mediating the effects of phenformin on both bacterial growth inhibition and B12 accumulation, we re-expressed the RcdA protein in the rcdA mutant strain. The complementary expression of RcdA nearly completely restored the bacterial sensitivity and B12-accumulating property in response to biguanide treatments (Fig. 4c, d). By using LC-MS/MS, we more directly measured the B12 levels in WT and ΔrcdA with or without drug treatment. Consistent with the results indicated by the B12 sensor animals (Fig. 4d), the deletion of rcdA significantly reduced the effect of phenformin in promoting B12 accumulation in both the Ado-Cbl and CN-Cbl forms compared to WT (Fig. 4e). We then examined the gene expression of rcdA and found it significantly increased upon treatments with both biguanides (Fig. 4f). Collectively, these results indicated that biguanides promote bacterial B12 accumulation by increasing the expression of rcdA.

To explore whether the deletion of rcdA abolished the effects of biguanides by modulating B12 transporters, we tested the expressions of B12 transporter genes in ΔrcdA with or without biguanide treatments. The results showed that neither metformin nor phenformin could upregulate the expression of those genes in ΔrcdA as they did in WT (Figs. 2d, 4g). Surprisingly, both biguanides instead even reduced the expression of btuB or btuD in ΔrcdA through a yet unknown mechanism (Fig. 4g). It implied that there might be a parallel pathway regulating the expressions of B12 transporter genes under biguanide treatments independent of RcdA. Together, our results suggested that biguanides elevate bacterial B12 uptake from the environment by promoting the expressions of B12 transporter genes, which presumably leads to B12 shortage in the environment, the host, over time (Supplementary Fig. 7).

## Discussion

Evident B12 deficiency has been routinely found to be associated with a range of symptoms, including diarrhea, anemia, and irreversible neuropathy[42,43]. More frequently than in other settings, B12 deficiency in patients taking metformin remains unrecognized, possibly due to its gradual development. Accordingly, there is no definitive guideline for the treatment of B12 deficiency in patients taking metformin thus far[44], despite B12 supplementation generally prescribed for relieving the possible resulting consequences. A better understanding of the mechanisms underlying metformin-induced B12 deficiency would contribute to the development of guidelines for the diagnosis and treatment of this specific type of B12 deficiency.

Using the model organism E. coli, our study demonstrated that metformin increases bacterial B12 accumulation from the environment. Through a combination of different screens with worms and E. coli, we mechanistically unveiled the molecular basis by which biguanide drugs exert their effects in helping bacterial B12 accumulation, i.e., by transcriptionally elevating the expressions of B12 transporter genes in an RcdA-dependent manner. Altogether, the results of our study provide a perspective for further investigating the action of metformin in inducing B12 deficiency in humans and a foundation for developing more targeted interventions against such side effects.

Metformin has been consistently suggested to be beneficial in promoting longevity and treating metabolic disorders in worms and humans through modulation of gut bacterial metabolism[16,18,45]. A recent study reported that metformin exerts immediate effects on changes of the composition and function of gut microbes in T2D patients. They found that the drug treatment is accompanied with a significant enrichment of genes responsible for bacterial environmental responses, such as ATP-binding-cassette transporters[16]. Other groups also found that metformin

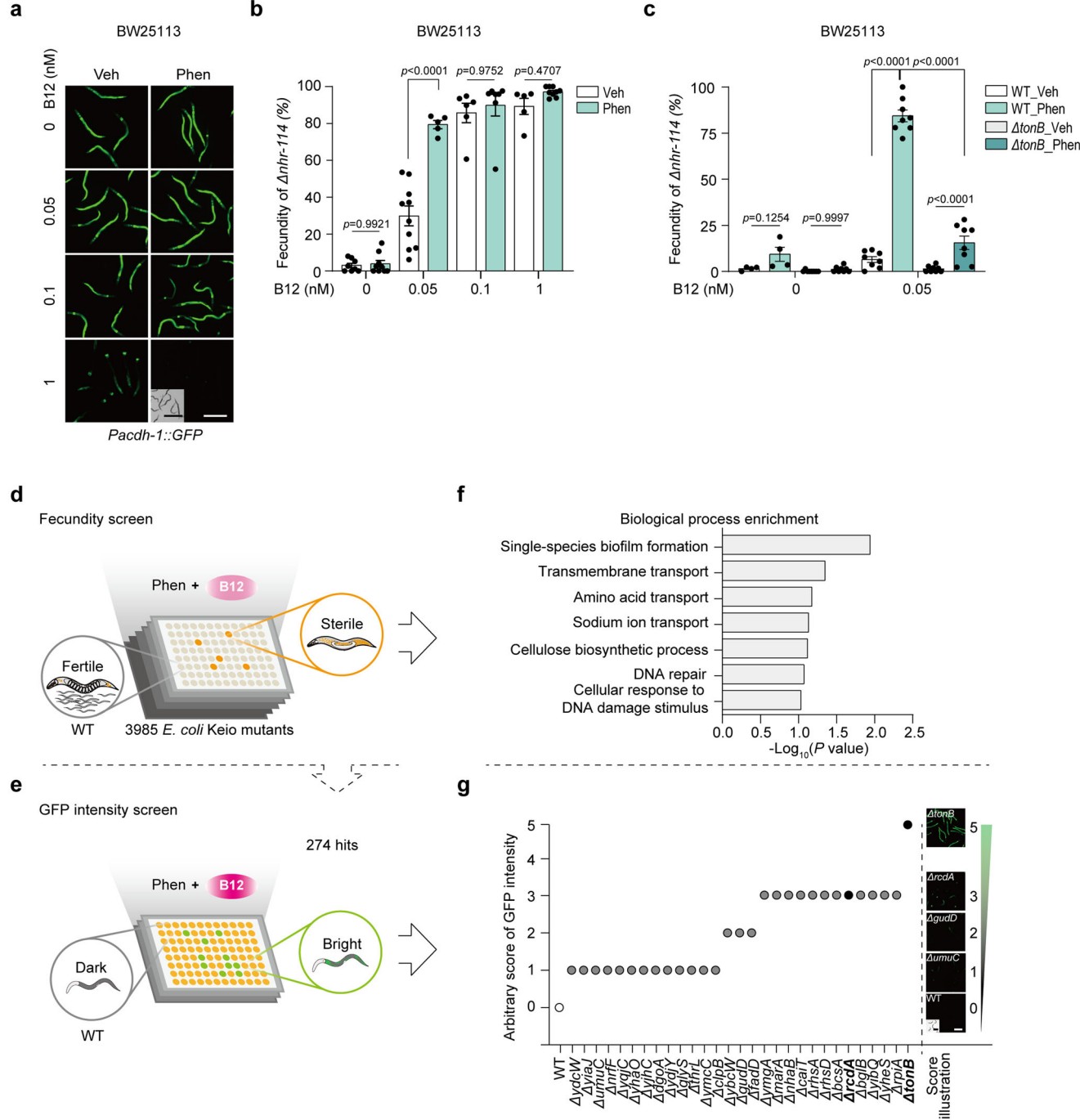

**Fig. 3 Bacterial genetic screens to identify the effectors of phenformin with worm B12 sensors. a** Representative images of *Pacdh-1::GFP* worms fed BW25113 pretreated with various doses of B12 and/or 4 mM phenformin. $N = 3$ independent experiments containing at least 30 worms per group. **b** The fecundity of *Δnhr-114* worms fed BW25113 pretreated with various doses of B12 and/or 2 mM phenformin. $N = 5$ independent experiments containing 1–2 replicates per experiment. **c** The fecundity of *Δnhr-114* worms fed WT and *ΔtonB* bacteria pretreated with 0.05 nM B12 and/or 2 mM phenformin. $N = 4$ independent experiments containing 1–2 replicates each time. The statistical significance values were determined by two-way ANOVA for (**b**, **c**). Error bars denoted the S.E.M. **d**, **e** Schemes of genetic screens for genes responsible for phenformin-mediated bacterial B12 accumulation. Fecundity screen (**d**). The cartoons of sterile and fertile *Δnhr-114* worms were adapted from our previous work[25]. The whole library was screened using *Δnhr-114* worms treated with 2 mM phenformin and 0.05 nM B12. $N = 4$ independent experiments containing at least 30 worms per well and each round. GFP intensity screen (**e**). A total of 274 hits from the fecundity screen were screened using *Pacdh-1::GFP* worms treated with 4 mM phenformin and 1 nM B12. $N = 3$ independent experiments containing at least 30 worms per well and each round. **f** Biological process enrichment analysis of 274 hits from the fecundity screen. **g** The fluorescence scores of *Pacdh-1::GFP* worms on WT and 28 hits from the GFP intensity screen. Scale bar: 250 μm (fluorescence images) and 500 μm (bright field images) for (**a**) and (**g**).

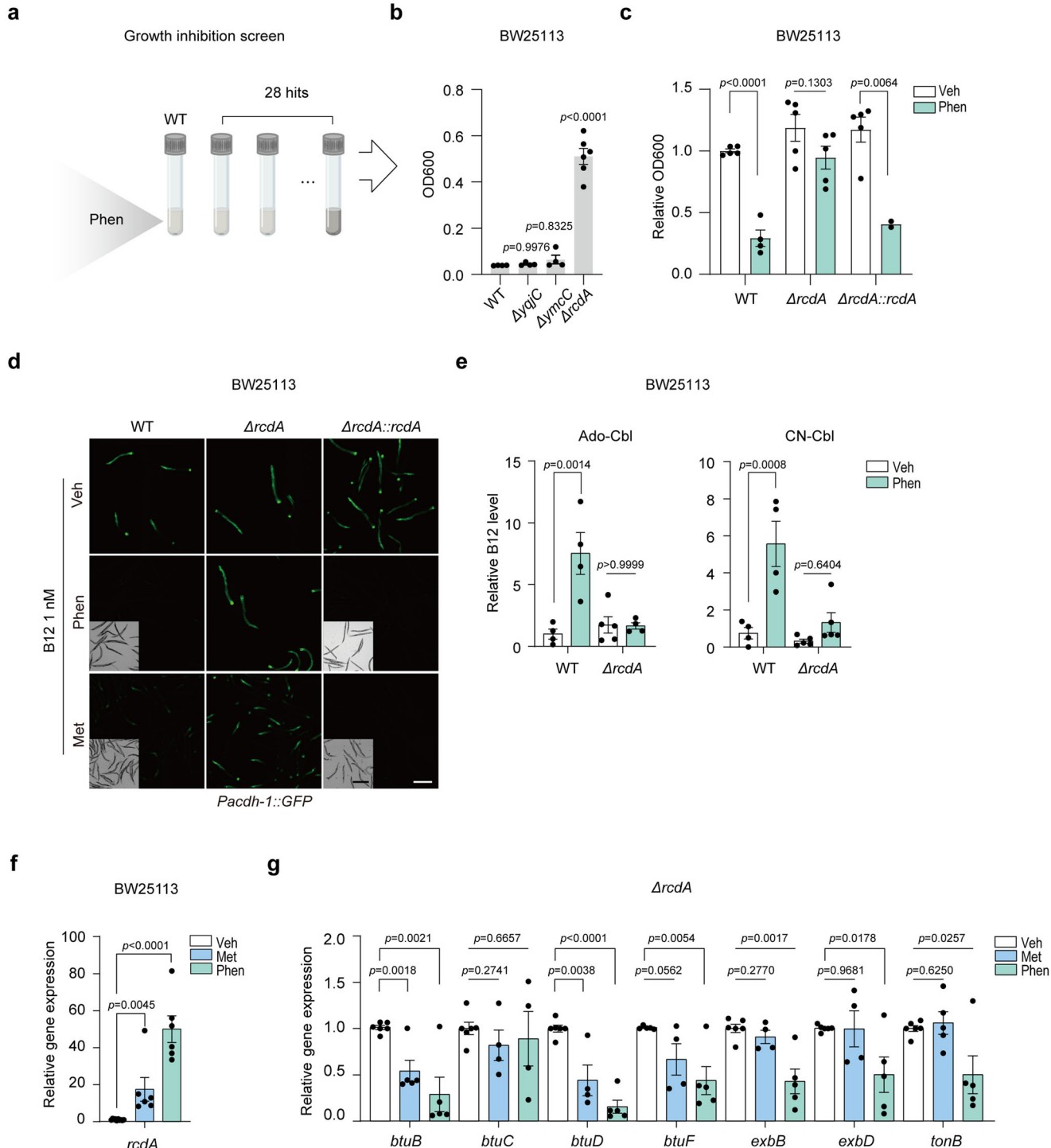

**Fig. 4 Biguanides induce bacterial growth inhibition and B12 accumulation through RcdA. a** Scheme of the growth inhibition screen. The growth statuses of 28 hits from the secondary screen after treatment with 5 mM phenformin were evaluated by optical density. **b** The outcomes of the growth inhibition screen. OD600 value measurement of WT, ΔyqjC, ΔymcC, and ΔrcdA. N = 2 independent experiments containing 6 replicates. **c** Growth of WT, ΔrcdA, and ΔrcdA::rcdA cells treated with 4 mM phenformin. N = 2 independent experiments containing 2–5 replicates. **d** Representative images of Pacdh-1::GFP worms fed WT, ΔrcdA, and ΔrcdA::rcdA pretreated with 4 mM phenformin/200 mM metformin or not. N = 3 independent experiments containing at least 30 worms per group. Scale bar: 250 μm (fluorescence images) and 500 μm (bright field images). **e** LC-MS/MS measurement of B12 levels in WT and ΔrcdA treated with 4 mM phenformin or untreated. N = 3 independent experiments containing 4–5 replicates. **f** Relative mRNA levels of rcdA with 200 mM metformin or 4 mM phenformin treatment. **g** Relative mRNA levels of genes involved in B12 transport in ΔrcdA with 200 mM metformin or 4 mM phenformin treatment. For (**f**, **g**), N = 2 independent experiments containing 4–6 replicates. The statistical significance values were determined by ordinary one-way ANOVA for (**b**) and (**e**) and multiple t tests for (**c**) and (**f**, **g**). Error bars denoted the S.E.M.

treatment can alter bacterial membrane function and activities in varied bacteria species[46,47]. There are a variety of bacteria colonizing the human distal ileum[48], the majority of which lack enzymes for *de novo* B12 biosynthesis and rely on the uptake of exogenous B12[49]. These bacteria have been reported with high capacities in obtaining B12 from the colonized host by modulating B12 transporters[50,51]. Here, we demonstrated that both biguanides could elevate B12 accumulation in multiple *E. coli* strains by increasing the expressions of B12 transporters in an RcdA-dependent manner. Whether biguanides can elicit such an effect in other species of gut microbiota warrants future studies.

The transcription factor RcdA is known with activities in mediating functions of the biofilm master regulator CsgD[28] and other players involved in stress responses[29]. Notably, the biofilm formation pathway was enriched in our initial screens, although depletions of those genes seemed no effects on modulating phenformin's growth inhibition activity in bacteria (Fig. 4b), indicating a potential role of biofilm formation in bacterial B12 gathering. RcdA is recently found in complex with multiple ligands likely to sense or respond to environmental cues[52]. Furthermore, our study proposed a previously unappreciated role of RcdA in mediating biguanide-induced bacterial B12 accumulation. The transport of B12 in *E. coli* is thought mainly relied on the Btu system and the energy-transducing TonB complex[53]. There are a number of ways reported to regulate the B12 transporter genes, including metal-dependent regulators, σ/anti-σ factor systems, small RNAs on TonB complex and the riboswitch regulation on *btuB*[54,55]. Whether and how biguanides modulate the expression of B12 transporter genes through these pathways, in a RcdA-dependent or -independent mode, warrant further investigations.

It is worth mentioning that this study presented an acute treatment model of metformin action using relatively high doses of the drug compared to its long-term and low-dose therapy in T2D patients. Our results showed that both metformin and phenformin can markedly promote bacterial B12 accumulation from the culture environment, but whether they elevate B12 accumulation in the physiological context is worthy of future research. Alternatively, a biguanide-induced B12-deficient mouse model, which is outside of our current expertise, can be applied in future studies to validate and advance our findings toward potential clinical applications.

## Methods

**Bacterial strains.** The *E. coli* strains OP50, HT115 and HB101 were obtained from the Caenorhabditis Genetics Center (CGC). The Keio Knockout Collection (Cat# OEC4988) and the parent strain BW25113 (Cat#OEC5042) were purchased from Dharmacon. *E. coli* strains were grown at 37 °C in normal LB or B12-deficient medium (tryptone replaced by neutralized soy-peptone, named soy medium thereafter). *E. coli* deletion mutants were grown at 37 °C in normal LB or soy medium with 50 μg/mL kanamycin[56].

**C. elegans strains.** Worms were maintained as described at 20 °C[57]. The strains VC1760 (*nhr-114(gk849)*) and VL749 (*wwIs24 [Pacdh-1::GFP + unc-119(+)]*) were purchased from the CGC. All experiments were performed with synchronized hermaphroditic animals.

**Chemicals.** CN-Cbl (Sigma Aldrich, Cat# V900445) was dissolved in ddH₂O as a stock solution at 15 mM. Metformin and phenformin (Sigma Aldrich, Cat# D150959 and Cat# PHR157) were prepared in ddH₂O as stock solutions at 1 M and 0.2 M, respectively. Varied doses of biguanides were applied in this study according to previous studies[18,38] and the level of tolerance of tested organisms to the drugs. In brief, 50 mM metformin was used in the on-plate treatment for *Δnhr-114* worms, and 200 mM was used in all bacterial treatments. The same volumes of ddH₂O were used as vehicle (Veh). Phenformin at 4 mM was used in all bacterial treatments for feeding *Pacdh-1::GFP* worms, LC-MS/MS measurement, growth measurement and qPCR. Phenformin was used at an optimized concentration, 2 mM, in the Keio Collection screen for feeding *Δnhr-114* worms to ensure sufficient bacterial growth. The highest dose of phenformin (5 mM) was used in on-plate treatments or growth inhibition assays to screen for the strongest

phenformin-resistant strains. All reagents were filtered with 0.22 μm membranes prior to use and stored at −20 °C.

**Fecundity assessment of Δnhr-114 worms.** *E. coli* OP50 was cultured in normal LB or soy medium and seeded on NGM or B12-deficient NGM (peptone was replaced by neutralized soya peptone, OXIOD, Cat# LP0044T) agar plates. The plates were treated with metformin and phenformin at the indicated concentrations and ddH₂O as Veh for 4 h before dropping the L1s of *Δnhr-114* worms. Alternatively, *E. coli* strains were cultured in soy medium with the indicated treatments at 37 °C for 12 h. The pretreated bacterial liquid was concentrated and seeded on B12-deficient NGM agar plates before dropping the L1s. The fecundity of the worms was quantified at the day 2 stage[58]. Animals were counted as fertile (with eggs in the uterus) or sterile (no eggs). The fecundity ratio was calculated as

$$\text{Fecundity ratio} = \text{the number of fertile worms} \div \text{the number of total worms} \times 100\%$$

(1)

**Imaging of Pacdh-1::GFP worms.** *E. coli* strains were cultured in soy medium with the indicated treatments at 37 °C for 12, 24 or 48 h. The pretreated bacterial liquid was concentrated and seeded on B12-deficient NGM agar plates. *Pacdh-1::GFP* worms in the L1 stage were dropped on the plates and cultured until the L4 stage. They were collected, anesthetized in M9 containing levamisole (1 mg/mL) and mounted on glass slides for imaging. All images were taken with a Leica DM500 microscope with fixed exposure parameters.

**Axenic medium culture.** The B12-deficient *C. elegans* Habituation and Reproduction (CeHR) medium was prepared using a recipe (Supplementary Data 2) modified from others[59]. Synchronized *Pacdh-1::GFP* worms in the L1 stage were cultured in CeHR medium supplemented with B12 at indicated concentrations with continuous shaking at 70 rpm on a shaker at 20 °C. Synchronization was conducted according to the hypochlorite method to remove bacteria.

**LC-MS/MS measurement of B12 and phenformin.** The levels of B12 and phenformin in *E. coli* were determined by LC-MS/MS. *E. coli* was cultured in 10 mL soy medium supplemented with 1 nM CN-Cbl and 4 mM phenformin or 200 mM metformin treatment for 12 h. Bacteria were collected by centrifugation and then washed with sterile M9 buffer 3 times. The bacteria were resuspended in 450 μL ddH₂O, and the samples were stored at −80 °C overnight. Samples were thawed and frozen with liquid nitrogen 3 times and lysed with the sonication device (Sonics VCX150 Ultrasonicator) using the program 10 s on 10 s off, for 5 mins, on ice. After sonication, the samples were centrifuged for 20 min to pellet the debris. The supernatant was transferred to new tubes, and the protein was precipitated with acetonitrile:methanol:supernatant (67.5:22.5:10 vol/vol/vol). The samples were then centrifuged at the maximum speed for 30 min at 4 °C. The supernatant was collected, lyophilized for 5 h, and resuspended in 50 μL ddH₂O for LC-MS/MS detection. The concentrations of B12 and phenformin were normalized to the total cell numbers.

LC-MS/MS analysis was conducted using an AB SCIEX QTRAP 6500+ mass spectrometer with an Exion LC system. Chromatographic separation was achieved on an ACQUITY UPLC BEH C18 column (100 mm × 2.1 mm, 1.7 μm) at 40 °C. The mobile phase consisted of water containing 0.1% formic acid, 20 mM ammonium acetate (A) and methanol (B) at a flow rate of 0.4 mL/min. The gradient of mobile phase B was 3% for 1 min, ramping up from 3% to 60% in 4 min, holding at 60% for 1 min, ramping up from 60% to 95% in 0.1 min, holding at 95% for 1.9 min, ramping down from 95% to 3% in 0.1 min, and holding at 3% for 1.9 min. The injected volume for each sample was 3 μL. The mass spectrometer was operated in positive ion mode using the following settings: curtain gas 40 psi, ion spray voltage 4.5 kV, source temperature 550 °C, ion source gas 1 60 psi, and ion source gas 2 55 psi. Compounds were measured by multiple reaction monitoring (MRM) with optimized cone voltage and collision energy.

**RNA extraction and quantitative RT–PCR.** The expressions of bacterial genes were evaluated by RT-qPCR. For gene expression evaluations, bacteria were inoculated in 3 mL regular LB medium at the ratio of 1% from overnight starters for 12 h. Specifically, for drug treated groups, bacteria were cultured in 3 mL soy medium supplemented with 1 nM CN-Cbl and 4 mM phenformin or 200 mM metformin for 12 h. Total RNA was extracted with RNAzol (GeneCopoeia, Cat# QP020) with bacterial pellets. gDNA was removed with DNase before reverse transcription with 5x Hiscript® III QRT SuperMix (Vazyme, Cat# R323-01). Quantitative PCR was conducted with SYBR Green PCR reagent (Vazyme, Cat# Q711-02) on a quantitative PCR system (Jena Qtower 3 G). For comparison in different bacterial strains, the expression levels of genes of interest were normalized to that of *idnT*, while those from the drug-treated groups were normalized to that of *rrsA* whose expression was not affected by biguanides. The sequences of the primers used are listed in Supplementary Data 3.

**E. coli deletion mutant screen with Δnhr-114 worms.** Keio *E. coli* deletion collection clones were grown overnight in 500 μL of soy medium containing 0.05 nM

B12, 2 mM phenformin, and 50 µg/mL kanamycin in 96-well deep-well plates. The overnight bacterial culture was concentrated to 25 µL (30 µL for those wells on the margin) and seeded onto B12-deficient 96-well plates. Approximately 30 synchronized L1 of Δnhr-114 animals were placed onto each well of the plates, and worms were allowed to develop for 96–120 h at 20 °C before observation. The fecundity phenotype in each well was scored from 0 (sterile) to 5 (fertile), as noted in Supplementary Data 1.

**Genotyping of bacterial deletion strains**. Single gene deletion bacterial strains were streaked onto LB plates containing 50 µg/mL kanamycin. PCR was carried out on individual colonies using genomic and kanamycin-cassette-specific primers. Genomic primers were designed for individual strains starting 100–1000 bases upstream of the start codon of the gene and 100–1000 bases downstream of the stop codon[60]. PCR products were analyzed for the correct sizes separated by agarose gel electrophoresis. The sequences of the primers used in this study were listed in Supplementary Data 3.

**Bacterial colony forming assay**. Bacterial viability was evaluated by comparing the colony forming units (C.F.U.) of bacterial cultures prior to and post phenformin treatment. The assay was performed with an initial bacterial culture containing $1 \times 10^6$ CFU bacteria/mL, which was diluted from overnight grown bacteria. The untreated sample was directly from the initial culture, and the phenformin-treated one obtained from the bacteria treated by the drug at 5 mM for another 12 h in 37 °C, with continuously shaking at 220 rpm. Both samples were diluted and streaked on LB agar plates followed by incubation at 37 °C for 12 h. The number of colonies formed were counted and the relative C.F.U. was calculated by

$$\text{Relative C.F.U.} = \text{drug treated C.F.U.}/(\text{Average of untreated C.F.U.}) \quad (2)$$

**Extracellular DNA (eDNA) staining by SYBR Green I**. Single colonies of BW25113 were inoculated and cultured in LB medium for 12 h as starters of the staining assay. These overnight cultures were aliquoted and treated with or without 5 mM phenformin for 6 or 12 h at 37 °C. Samples from each time point were collected for OD600 measurement and stained with SYBR Green I (MCE, Cat# HY-K1004, 10,000x). Images of bacteria post SYBR Green I staining were acquired with Olympus FV3000-BX63 with 63x immersion objective and analyzed by ImageJ software. More than 3 images per slide were taken for fluorescence quantification.

**Bacterial growth inhibition assay**. Bacteria were cultured in 200 µL LB supplemented with 5 mM phenformin in 96-well flat-bottom plates at 950 rpm for 12 h. The optical density (OD) of the overnight bacterial culture was obtained by using a plate reader (Thermo Varioskan LUX Microplate reader) at a wavelength of 600 nm. The experiment was performed in duplicate and repeated two times.

**Construction of the complementary strain**. To construct the plasmids for ΔrcdA complementation, the rcdA fragment was amplified and then ligated into the pUC57::amp plasmid (Addgene#196258). The generated recombinant vector (pUC57::amp::rcdA) was subsequently transformed into the competent ΔrcdA mutant. The complementation strain loaded with pUC57::amp::rcdA was selected on an LB plate containing ampicillin. Finally, the selected complementation strain was validated by PCR and designated ΔrcdA::rcdA.

**Statistics and reproducibility**. GraphPad Prism 8.0 (GraphPad Software, Inc.) was used for statistical analyses in this study. All results were obtained from at least three biologically replicates. For experiments using C. elegans, at least 30 worms were analyzed for each condition. For differences between two groups, unpaired Student's t test was used. For differences among multiple groups, one-way analysis of variance (ANOVA) followed by Tukey's test was used. Two-way ANOVA was used to analyze differences between multiple groups with two variations. $p < 0.05$ was considered to be statistically significant. Data are presented as the mean ± S.E.M. The statistical significance of differences was inserted in each figure.

**Reporting summary**. Further information on research design is available in the Nature Portfolio Reporting Summary linked to this article.

## Data availability

The authors declare that all data of this study have been provided within the article and Supplementary data set. The source data for all presented figures were provided in Supplementary Data 4. Further information, resources, and reagents are available from the corresponding author upon reasonable request.

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

## Acknowledgements

The authors thank Jinheng Pan, Jia Chen, Shan Feng, Zhenzhen Yu, and Yalin Wang for facility support and the *C. elegans* Genetics Center (CGC) for providing strains. This work was supported by the Westlake Education Foundation and the National Natural Science Foundation of China (32071151).

## Author contributions

L.W., L.Y., and Y.W. together conceived and designed the study. L.Y. and Y.W. both conducted experiments and data analysis and wrote a draft of the work; S.Q. contributed to experiments with Δ*nhr-114* animals and related discussions. S.Z. contributed to the axenic culture experiments. L.W. supervised the project and wrote the manuscript with input from all authors.

## Competing interests

The authors declare no competing interests.
