## [Peer Review File · Communications Biology]

Reviewers' comments:

Reviewer #1 (Remarks to the Author):

In this manuscript, Yao et al. employed functional and genomic approaches to examine whether and how the antidiabetic drug metformin facilitates B12 accumulation of bacteria from the environment. Results showed that both metformin and the other biguanide drug phenformin elevated B12 accumulation in *E. coli* and increased the expression of B12 transport genes. Besides, the transcription factor RcdA was identified as an element responsible for the B12 accumulation. The authors concluded that metformin helps bacteria gather B12 from the environment by increasing the expression of B12 transporter genes in an RcdA-dependent manner.

This work is comprehensive and interesting. The conclusions are novel and supported by data. This work provides important mechanistic insights into the effect of biguanide drugs in helping bacteria capture B12 from the environment and the long-observed association between metformin use and B12 deficiency in type 2 diabetes patients. Besides, the manuscript is clearly and concisely written. A major limitation of this work lies in the fact that a majority of the mechanistic exploration experiments were conducted solely using phenformin, the stronger version of metformin. Nonetheless, metformin was included in all crucial experiments which directly led to the major conclusions.

The discussion section is a major area of improvement for this manuscript. The current discussion is short and largely repeats the contents in the introduction and results. The authors should try to discuss their findings in the context of previous literature. Specifically, the transcription factor RcdA and its potential link to the B12 transporter system should be elaborated.

Other concerns:

- P5 L89: it is the tonB gene which was knocked out but not the pathways
- P8 L158: in fact, only 4 out of 7 of the B12 transporter genes showed significant results
- P8 L161: I think the authors mean "although not for the other tested genes from the B12 transport system".
- P9 L178: although not statistically significant, there was actually some noticeable increase in the delta tonB group with B12 and Phen when compared to the WT group. Is it possible that the insignificant result was actually due to a small sample size (N = 4) used?
- P11 L225: Figure 2d instead of 2e
- P11 L226: is there any possible explanation for the reduction? Also, what does the reduction imply?
- P29 Figure 1d: provide the full names of the two B12 forms in the legend
- P35 Figure 4b: how about the other 25 targets? Was there any reason why delta yqjC and ymcC were chosen here?
- P39 Figure S3b: this figure should be moved to the main text since it shows the effect of metformin, the main target of the study, on the expression of B12 transporter genes.

Reviewer #2 (Remarks to the Author):

Using the *C. elegans* model, Yao et al. present a study describing their finding that the biguanide drug helps *E. coli* gather vitamin B12 from the environment by increasing the expression of VB12 transporter genes in a RcdA-dependent manner. The result is attention-grabbing; it may help to explain VB12 deficiency in T2D patients with long-term use of the antidiabetic drug metformin. The authors did a very good job of presenting their findings, making the manuscript easy to follow. The study was well-done, but some experiments could be articulated more clearly. Here are my concerns -

1. Before talking about the results in Fig 1b, it would be helpful if the authors could provide some background information about the VB12-sensing GFP worms.

2. The commonly used *E. coli* strain for *C. elegans* is OP50, and the authors used OP50 in their first experiment – Fig 1a. But in Fig 1c, they switched to another *E. coli* strain, BW25113. Different *E. coli* strains were discussed later, but some explanation is needed here.

3. It would be helpful if the authors could draw simple sketches to elucidate the different treatments of the worms and add them along with Fig 1b and 1c. Based on the description in M&M, my understanding is that in Fig 1b, worms were synchronized in the VB12-deficient CeHR medium, and then L1 stage worms were dropped onto CeHR agar plates which contain 0 or 15 nM VB12, and Veh or Phen; while in Fig 1c, *E. coli* were incubated with 0 or 1 nM VB12, and with Veh or Phen, for 12 hr, and then dropped onto VB12-deficient NGM plates, and then L1 stage worms were transferred onto the plates. It may be hard to follow for those who have never worked with *C. elegans*.

4. Could the authors justify the concentrations of metformin and phenformin used in the study? In the fecundity experiment, they used 50 mM metformin and 5 mM phenformin, but in Fig 1c, they used 200 mM metformin and 4 mM phenformin. And how is the concentration used in this study compared to the actual physiological drug level in T2D patients who have been using the drug for an extended period?

5. In Fig 2d, qPCR was performed to measure the expression level of VB12 transport genes +/-4 mM Phen. It's not clear to me how long the bacteria were treated with Phen, was Phen added to the growth medium from the beginning? The authors later showed that 5 mM Phen dramatically inhibited bacteria growth (Fig 4b). Would 4mM Phen cause any growth inhibition?

6. In the growth inhibition assay, 5 mM Phen was added to the medium during inoculation. What is the mode of action of phenformin, bacteriostatic or bactericidal? If 5 mM Phen were added to a log phase or an overnight *E. coli* culture, would you see any growth inhibition or cell lysis? And, is the growth inhibition strain-dependent (what about OP50)? This is also related to my question 4 – “how is the concentration used in this study compared to the actual physiological drug level in T2D patients who have been using the drug for an extended period ?” If the drug is bactericidal, even if it can stimulate the uptake of VB12 by bacteria, VB12 will eventually be released into the environment due to cell lysis. If the drug is bacteriostatic, the bacterial number will not increase, and the amount of VB12 uptake will reach a plateau, which would not cause a long-term VB12 deficiency.

Reviewer #3 (Remarks to the Author):

The manuscript by Luxia Yao et al. estimates an effect of biguanides on the ability of *E. coli* to increase the accumulation of vitamin B12. This is a carefully designed and well performed study that has led to the identification of the mechanism responsible for the accumulation of B12 in *E. coli*. The paper is well-written and displays a set of elegant experiments containing the necessary controls to validate the conclusions.

Few minor comments:

Details on the control (vehicle) experiments should be clarified in the Methods (e.g. was the water used instead of Met or Phen solutions and how they were applied to the plates).

Why the Ado-CBL and CN-Cbl concentrations were only estimated for the BW25113 strain and not the other *E. coli* strains used in experiments?

Also, the potential explanation for the difference between relative levels of Ado-Cbl and CN -CBL (Figure 1d) depending on either the Met or Phen treatment is not discussed. (in other words: what can

cause a higher proportion of Ado-Cbl after the Met treatment and higher CN-Cbl after the Phen treatment?)

Is there a reason why BW25113 is not added to the comparative gene expression in Figure 2c ?

Point-by-point response to reviewers, COMMSBIO-22-3232A

Reviewer #1 comment 1: ...This work is comprehensive and interesting. The conclusions are novel and supported by data. This work provides important mechanistic insights into the effect of biguanide drugs in helping bacteria capture B12 from the environment and the long-observed association between metformin use and B12 deficiency in type 2 diabetes patients. Besides, the manuscript is clearly and concisely written.

RESPONSE: We sincerely appreciate the reviewer for finding our work interesting and well conveyed.

Reviewer #1 comment 2: A major limitation of this work lies in the fact that a majority of the mechanistic exploration experiments were conducted solely using phenformin, the stronger version of metformin. Nonetheless, metformin was included in all crucial experiments which directly led to the major conclusions.

RESPONSE: We thank the reviewer for this important criticism and the recognition that metformin has been included in all crucial experiments. Metformin is costly for high throughput screens, especially when high doses of the drug are needed. Therefore, we used the stronger version of biguanides phenformin rather than metformin in our screens. To address this specific concern as much as possible, we have added metformin test in the *rcdA* mutant and its complementary strains. The results showing consistency of mechanisms for action of both biguanides have been updated in the revised

manuscript (Main text Fig. 4d, Page 11 Lines 221-222).

Reviewer #1 comment 3: The discussion section is a major area of improvement for this manuscript. The current discussion is short and largely repeats the contents in the introduction and results. The authors should try to discuss their findings in the context of previous literature. Specifically, the transcription factor RcdA and its potential link to the B12 transporter system should be elaborated.

RESPONSE: This is an insightful point. We have implemented more discussions as suggested mainly including a summary of metformin effect on microbes in humans, a specific section for RcdA study, the regulation of B12 transporter system and limitations of our *in vitro* model (Pages 13-15 Lines 263-300).

Reviewer #1 comment 4: Other concerns: P5 L89: it is the tonB gene which was knocked out but not the pathways

RESPONSE: We apologize for the confusion and have mended the wording in the revised version (Page 5 Line 87)

Reviewer #1 comment 5: P8 L158: in fact, only 4 out of 7 of the B12 transporter genes showed significant results

RESPONSE: The reviewer is correct. We have revised the statement accordingly (Page 8 Line 167).

Reviewer #1 comment 6: P8 L161: I think the authors mean "although not for the other tested genes from the B12 transport system".

RESPONSE: We apologize for the oversight and have implemented the reviewer's suggestion (Page 9 Line 170).

Reviewer #1 comment 7: P9 L178: although not statistically significant, there was actually some noticeable increase in the delta tonB group with B12 and Phen when compared to the WT group. Is it possible that the insignificant result was actually due to a small sample size (N = 4) used?

RESPONSE: This point is insightful. We have repeated the mentioned experiment (total N=8). As predicted by the reviewer, we found a significant increase of the fecundity of *Δnhr-114* animals in the delta *tonB* group with B12 and Phen, although the increment is subtle compared to that of WT (Main text Fig. 3c, Page 9 line 187).

Reviewer #1 comment 8: P11 L225: Figure 2d instead of 2e

RESPONSE: We apologize for the oversight and have corrected the typo (Page 12 Line 235).

Reviewer #1 comment 9: P11 L226: is there any possible explanation for the reduction? Also, what does the reduction imply?

RESPONSE: We thank the reviewer for close analysis of our data and have also noticed the mentioned reduction at the time of seeing the results. Frankly, we have not

thought out any explanation of perfection for such information. It may indicate a parallel pathway along with RcdA for the regulation of B12 transporter expression in response to the drug treatment. A related speculation has been added in the revised manuscript (Page 12 Lines 235-238).

Reviewer #1 comment 10: P29 Figure 1d: provide the full names of the two B12 forms in the legend

RESPONSE: Done (Page 31 Line 644).

Reviewer #1 comment 11: P35 Figure 4b: how about the other 25 targets? Was there any reason why delta yqjC and ymcC were chosen here?

RESPONSE: We apologize for the confusion and have made it clearer now (Page 10 Lines 206-208). Out of the 28 hits from our GFP intensity screen, only those presented three mutants ($\Delta yqjC$, $\Delta ymcC$ and $\Delta rcdA$) can grow in the culture with 5 mM phenformin. Thus, we only displayed the successfully grown ones in the mentioned figure.

Reviewer #1 comment 12: P39 Figure S3b: this figure should be moved to the main text since it shows the effect of metformin, the main target of the study, on the expression of B12 transporter genes.

RESPONSE: This advice has been implemented in the revised version (Main text Fig. 2d, Page 8 Lines 166-168).

Reviewer #2 comment 1: ...The result is attention-grabbing; it may help to explain VB12 deficiency in T2D patients with long-term use of the antidiabetic drug metformin. The authors did a very good job of presenting their findings, making the manuscript easy to follow. The study was well-done, but some experiments could be articulated more clearly.

RESPONSE: Thanks a lot for the reviewer's recognition of the readability of our manuscript and the significance of our work. We now have tried as much as we can to articulate the suggested experiments in the revised version.

Reviewer #2 comment 2: Here are my concerns -Before talking about the results in Fig 1b, it would be helpful if the authors could provide some background information about the VB12-sensing GFP worms.

RESPONSE: This helpful suggestion has been implemented in the revised text (Page 6 Lines 109-110).

Reviewer #2 comment 3: The commonly used E. coli strain for C. elegans is OP50, and the authors used OP50 in their first experiment – Fig 1a. But in Fig 1c, they switched to another E. coli strain, BW25113. Different E. coli strains were discussed later, but some explanation is needed here.

RESPONSE: This point is well made. We have added the suggested explanation in the revised manuscript (Page 6 Line 113).

Reviewer #2 comment 4: It would be helpful if the authors could draw simple sketches to elucidate the different treatments of the worms and add them along with Fig 1b and 1c. Based on the description in M&M, my understanding is that in Fig 1b, worms were synchronized in the VB12-deficient CeHR medium, and then L1 stage worms were dropped onto CeHR agar plates which contain 0 or 15 nM VB12, and Veh or Phen; while in Fig 1c, E. coli were incubated with 0 or 1 nM VB12, and with Veh or Phen, for 12 hr, and then dropped onto VB12-deficient NGM plates, and then L1 stage worms were transferred onto the plates. It may be hard to follow for those who have never worked with C. elegans.

RESPONSE: We appreciate the reviewer's thoughtful suggestion and have added the suggested schemes to revised figures (Main text Fig. 1b, c).

Reviewer #2 comment 5: Could the authors justify the concentrations of metformin and phenformin used in the study? In the fecundity experiment, they used 50 mM metformin and 5 mM phenformin, but in Fig 1c, they used 200 mM metformin and 4 mM phenformin.

RESPONSE: We apologize for the confusion. Detailed explanations have been added in the revised method section (Pages 15-16 Lines 319-325).

Reviewer #2 comment 6: And how is the concentration used in this study compared to the actual physiological drug level in T2D patients who have been using the drug for an extended period?

RESPONSE: This is an outstanding question. Our study applied high doses of biguanides but for a relatively much shorter time in the *in vitro* experiments, compared to the drug treatment in T2D patients. According to a previous study, metformin level in T2D patients is estimated as 500 ug/mg tissue (3 mM) in the gut (Bailey *et al.*, 2008; McCreight *et al.*, 2016). Notably, B12 deficiency is commonly observed in patients with long-term metformin treatment, occasionally post years of usage of the drug (de Jager *et al.*, 2010; Chapman *et al.*, 2016). A meaningful note has been added in the revised discussion section (Page 14 Lines 293-295).

Reviewer #2 comment 7: In Fig 2d, qPCR was performed to measure the expression level of VB12 transport genes +/-4 mM Phen. It's not clear to me how long the bacteria were treated with Phen, was Phen added to the growth medium from the beginning?

RESPONSE: We apologize for the confusion. Exactly as the reviewer predicted, phenformin was added to the growth medium at the beginning of the culture. The detailed information has been added in the method section (Page 19 Lines 390-395).

Reviewer #2 comment 8: The authors later showed that 5 mM Phen dramatically inhibited bacteria growth (Fig 4b). Would 4mM Phen cause any growth inhibition?

RESPONSE: Phenformin at 4 mM also restricted the growth of bacteria but not as strong as it does at 5 mM, which was used in the screen for the strongest phenformin-resistant strains (Main text Fig 4b and 4c, and also see Response to *Reviewer #2*

comment 11). We have emphasized this point in the revised method section (Page 16 Lines 323-325).

Reviewer #2 comment 9: *In the growth inhibition assay, 5 mM Phen was added to the medium during inoculation. What is the mode of action of phenformin, bacteriostatic or bactericidal?*

RESPONSE: To answer this specific question, we performed the colony-forming assay with bacteria treated with 5 mM Phen or not according to a modified protocol from the previous study by Balouiri *et al* (Balouiri *et al.*, 2016). The results showed no significant differences in colony-forming units between the two groups despite a reduction trend present in the phenformin-treated group (Response Fig. 1), indicating that the action of the drug at 5 mM should be bacteriostatic.

Response Fig. 1 Colony-forming assay with BW25113 bacteria treated by 5 mM Phen or not.

Reviewer #2 comment 10: *If 5 mM Phen were added to a log phase or an overnight E. coli culture, would you see any growth inhibition or cell lysis?*

RESPONSE: We conducted the suggested experiments and found that 5 mM Phen indeed induced dual effects on growth inhibition measured by OD600 and cell lysis indicated by the dsDNA binding dye SYBR Green I (Response Fig. 2a and 2b). It was also noted there were only a small proportion of bacteria affected by the over cultivation with or without the high-dose drug treatment according to the SYBR staining results (Response Fig. 2b). As stated in the responses to *Reviewer #2 comment 6*, a much lower metformin dose is used in the physiological context in T2D patients where the drug may not induce strong effects on either growth inhibition or cell death in bacteria.

Response Fig. 2 Status of overnight-cultured bacteria in response to a 5 mM Phen treatment for 12 hours.

Reviewer #2 comment 11: And, is the growth inhibition strain-dependent (what about OP50)?

RESPONSE: We have explored this question by measuring OD600 values of different *E. coli* strains treated with 4 mM phenformin. The results showed that phenformin retarded the growth of all four tested *E. coli* strains despite varying degrees of inhibition (Response Fig. 3), suggesting that the growth inhibition activity of phenformin is not

strain-dependent.

Response Fig. 3. Growth measurement in varied bacterial strains treated by 4 mM phenformin or not.

Reviewer #2 comment 12: This is also related to my question 4 – “how is the concentration used in this study compared to the actual physiological drug level in T2D patients who have been using the drug for an extended period ?” If the drug is bactericidal, even if it can stimulate the uptake of VB12 by bacteria, VB12 will eventually be released into the environment due to cell lysis. If the drug is bacteriostatic, the bacterial number will not increase, and the amount of VB12 uptake will reach a plateau, which would not cause a long-term VB12 deficiency.

RESPONSE: This is a thoughtful comment. We certainly agree with the reviewer that B12 deficiency would not be induced if a large proportion of bacteria is killed or retarded by metformin. In fact, most of our results were obtained with an acute treatment (a much shorter time) and a much higher dose of the drug than that applied in T2D patients. Instead, numerous studies have reported that clinical doses of metformin usage can increase the abundance of *E. coli* and decrease *Intestinibacter*

bartlettii in humans (Bryrup *et al.*, 2019; Elbere *et al.*, 2018; Forslund *et al.*, 2015; Sun *et al.*, 2018; Vich Vila *et al.*, 2020; Wu *et al.*, 2017). However, when considering a lower concentration and longer course of treatment *in vivo*, whether and how the drug can induce bacterial B12 accumulation still needs further exploration (Page 14 Lines 293-295). A corresponding result from experiments with a lower dose and longer treatment of phenformin and related discussions have been updated in the revised version (Main text Supplementary Fig. 1, Page 6 Lines 116-123).

Reviewer #3 comment 1: The manuscript by Luxia Yao et al. estimates an effect of biguanides on the ability of E. coli to increase the accumulation of vitamin B12. This is a carefully designed and well performed study that has led to the identification of the mechanism responsible for the accumulation of B12 in E. coli. The paper is well-written and displays a set of elegant experiments containing the necessary controls to validate the conclusions.

RESPONSE: We appreciate that the reviewer finds our study well-designed and the manuscript well-written.

Reviewer #3 comment 2: Few minor comments: Details on the control (vehicle) experiments should be clarified in the Methods (e.g. was the water used instead of Met or Phen solutions and how they were applied to the plates).

RESPONSE: The suggestion has been implemented in our revised methods section (Page 15 Lines 323 and 336-337).

Reviewer #3 comment 3: Why the Ado-CBL and CN-Cbl concentrations were only estimated for the BW25113 strain and not the other E. coli strains used in experiments?

RESPONSE: We thank the reviewer for this important point and have included all other strains mentioned here in the revised experiments (Main text Supplementary Fig. 2a, Pages 7-8 Lines 143-146).

Reviewer #3 comment 4: Also, the potential explanation for the difference between relative levels of Ado-Cbl and CN -CBL (Figure 1d) depending on either the Met or Phen treatment is not discussed. (in other words: what can cause a higher proportion of Ado-Cbl after the Met treatment and higher CN-Cbl after the Phen treatment?)

RESPONSE: We thank the reviewer for the careful analysis of our data and the insightful question. A meaningful discussion has been added into the revised text (Page 7 Lines 129-135). In brief, we speculate that the more potent Phen may slow down the bacterial transformation of Ado-Cbl from CN-Cbl compared to Met, and thus yield a higher proportion of CN-Cbl than Met does.

Reviewer #3 comment 5: Is there a reason why BW25113 is not added to the comparative gene expression in Figure 2c?

RESPONSE: This point is well made. We also feel that it is important to include the most used BW25113 in our study into the mentioned experiment, and have done this accordingly (Main text Fig. 2c, Page 8 Lines 160-161).

REVIEWERS' COMMENTS:

Reviewer #1 (Remarks to the Author):

The authors have addressed all my previous concerns in this revised manuscript.

Reviewer #3 (Remarks to the Author):

After reviewing the answers provided by the authors to the previously made comments and suggestions I conclude that all raised questions have been answered and substantial improvements have been made to the manuscript by introducing the necessary amendments.